# The geographical maldistribution of obstetricians and gynecologists in Japan

**Kunichika Matsumoto[1], Kanako Seto[1], Eijiro Hayata[2], Shigeru Fujita[1], Yosuke Hatakeyama[1], Ryo Onishi[1], Tomonori Hasegawa[1]***

**1** Department of Social Medicine, Toho University School of Medicine, Tokyo, Japan, **2** Department of Obstetrics and Gynecology, Toho University Omori Medical Center, Tokyo, Japan

* tommie@med.toho-u.ac.jp

## Abstract

### Background

In Japan, there is a large geographical maldistribution of obstetricians/gynecologists, with a high proportion of females. This study seeks to clarify how the increase in the proportion of female physicians affects the geographical maldistribution of obstetrics/gynecologists.

### Methods

Governmental data of the Survey of Physicians, Dentists and Pharmacists between 1996 and 2016 were used. The Gini coefficient was used to measure the geographical maldistribution. We divided obstetricians/gynecologists into four groups based on age and gender: males under 40 years, females under 40 years, males aged 40 years and above, and females aged 40 years and above, and the time trend of the maldistribution and contribution of each group was evaluated.

### Results

The maldistribution of obstetricians/gynecologists was found to be worse during the study period, with the Gini coefficient exceeding 0.400 in 2016. The contribution ratios of female physicians to the deterioration of geographical maldistribution have been increasing for those under 40 years and those aged 40 years and above. However, there was a continuous decrease in the Gini coefficient of the two groups.

### Conclusions

The increase in the contribution ratio of the female physician groups to the Gini coefficient in obstetrics/gynecology may be due to the increased weight of these groups. The Gini coefficients of the female groups were also found to be on a decline. Although this may be because the working environment for female physicians improved or more female physicians established their practice in previously underserved areas, such a notion needs to be investigated in a follow-up study.

**Data Availability Statement:** Most of Data are available at: https://www.e-stat.go.jp/stat-search?page=1&toukei=00450026&kikan=00450 However, sharing a de-identified data set requires

permission from the Ministry of Health, Labour and Welfare based on statistical methods (https://www.mhlw.go.jp/toukei/itaku/tokumei.html). Under the permission of the Ministry of Health, Labour and Welfare (Seito-0408-1), we obtained anonymized individual data from this census. Therefore, we cannot share de-identified data sets and interim aggregated data sets used for this analysis.

**Funding:** This study was supported in part by the Japan Society for the Promotion of Science KAKENHI Grant No. 18K09955. There is no role of funder in this study. There was no additional external funding received for this study.

**Competing interests:** The authors have declared that no competing interests exist.

## Introduction

In Japan, the sense of physician shortage worsened in the 2000s, becoming a social concern at the peak of 2008 [1]. Although reportage of the shortage has lessened in the media, the maldistribution of physicians by regions and medical specialties remains a challenge. Recently, the overwork of Japanese workers has become a problem, and work style reforms are being made by the Japanese government. Physicians have been required to improve their work styles, and a study group on reforming physicians' work styles established by the Ministry of Health, Labor and Welfare (MHLW), issued a report in 2019 [2]. The report requires an upper limit on working hours, raising concerns about its possible impact on the current medical system in areas and specialties with physician shortage.

Regarding obstetricians/gynecologists, the number of physicians has not increased so much in recent years compared to other specialties, though the proportion of female physicians has increased. However, the geographical maldistribution has not been resolved [3]. In addition, overtime work in obstetrics/gynecology is longer than other specialties, suggesting the need for further improvement of the working environment [4]. If the working environment of obstetricians/gynecologists does not improve, it would be difficult to attract young physicians into the specialty. Such a scenario will further deteriorate the shortage problem, with untoward consequences for the medical care system [3, 5]. The Japan Association of Obstetricians and Gynecologists has been conducting a yearly questionnaire survey since 2007 in hospitals that handle deliveries [4]. It reported that (1) the number of deliveries decreased, but the number of high-risk cases increased, though this was hardly suggestive of a decline in the demand for obstetric and gynecological services, (2) among full-time physicians, the number of male physicians decreased, and the number of female physicians who were pregnant or raising children increased, (3) the working environment regarding female physicians who were pregnant or raising children were gradually improving, but it could not be said to be sufficient, (4) female freelance physicians concentrated in metropolitan areas and about half of them were in their 30s and 40s. The supply and demand imbalance seems to be largely unchanged, with new challenges associated with an increase in the number of female physicians.

Previous studies have addressed the supply and demand as well as the geographical maldistribution of physicians in Japan [6–13], but there only few studies have empirically analyzed the data on the situation of obstetricians/gynecologists [1, 3, 14]. Furthermore, regarding geographical maldistribution of obstetricians/gynecologists, there is only one estimation of the index of geographical physician maldistribution by the Study Group on the Supply and Demand of Medical Workers of the MHLW [15]. Therefore, there is a dearth of knowledge about how the geographical maldistribution has changed over time, and the possible causes of that change.

Therefore, this study investigated the time-series changes in the geographical maldistribution of obstetricians/gynecologists, analyzing the influence of the increase in the proportion of female physicians on their geographical maldistribution.

## Data sources and methods

### Data sources

In Japan, physicians need to report their workplace and specialty to the MHLW every two years. The MHLW has been publishing data on the number of physicians in each municipality as the Survey of Physicians, Dentists, and Pharmacists since 1972. This survey is the sole source of data that comprehensively captures the number of physicians in Japan. The present study used the Survey of Physicians, Dentists and Pharmacists between 1996 to 2016; the period for

which computerized individual data was available. Under the permission of the Ministry of Health, Labour and Welfare (Seito-0408-1), we obtained anonymized individual data from this census.

To determine the population of municipalities by 5-year age groups, we used data from basic resident registers compiled and published by the government. The population data from basic resident registers is inferior in terms of expressing reality compared to the national census, which is conducted by distributing survey forms to actual residents. However, the census is conducted only once every five years, whereas the population data from basic resident registers use annual data. In this study, we used the data by municipality, sex, and 5-years-age group.

## Methods

### Current situation of the geographical maldistribution

In order to measure maldistribution among municipalities, it is necessary to align the municipalities of the data for each year. In Japan, many municipalities merged between 1996 and 2016. Therefore, we readjusted the numbers for the physicians and population in each year according to the 2010 boundaries. The total number of municipalities was 1,865.

The Gini coefficient has been widely used to measure the geographical maldistribution of physicians [6–11]. It is a coefficient indicating inequality and has a value that ranges from 0 to 1. The closer to 1 the value is, the more unequal and the higher the maldistribution of physicians. In this study, the Gini coefficient of obstetricians/gynecologists was calculated in time series while that of all physicians and pediatricians were also calculated for comparison. As the relative frequency in the calculation process, the total population of the municipality was used for the Gini coefficient of all physicians, 15-49-year-old female population was used for that of obstetricians/gynecologists, and 0-14-years-old population was used for that of pediatricians. In this study, the Gini coefficient was calculated from 1996 to 2018.

### Increase of female obstetricians/gynecologists and geographical maldistribution

One of the characteristics of obstetricians/gynecologists is that the proportion of females is higher than those of other specialties. It is also known that the percentage of young female physicians who choose obstetrics/gynecology has been increasing year by year. We analyzed how the increasing proportion of female physicians affected the geographical maldistribution by decomposing the Gini coefficient. For analysis, we first divided obstetricians/gynecologists into four groups based on age and gender: males under 40 years, females under 40 years, males aged 40 years and above, and females aged 40 years and above. The reason for using 40 years as the cut-off is that at this, a physician would have had about 10 years of clinical experience after postgraduate clinical training and begins to play major clinical roles. The obstetricians/gynecologists in each group may also face the problem of geographical maldistribution. We decomposed the Gini coefficient by groups using Rao's method [16, 17]. When G is the overall Gini coefficient for obstetricians/gynecologists in municipalities, we can decompose G as follows:

$$G = \sum_{i=1}^{m} W_i \cdot \bar{G}_i$$

where i is the each group (males under 40 years, females under 40 years, males aged 40 years and above, and females aged 40 years and above), $W_i$ is the proportion of the number of

obstetricians/gynecologists who belong to group i, and $\bar{G}_i$ is the pseudo-Gini coefficient (the numerical value obtained if municipalities are arrayed in increasing order of total obstetricians/gynecologists per capita, rather than in increasing order of obstetricians/gynecologists per capita of the group.). $W_i \cdot \bar{G}_i$ is the weighted pseudo-Gini coefficient, and proportion of $W_i \cdot \bar{G}_i$ to G shows the contribution ratio for the maldistribution of obstetricians/gynecologists in group i. In addition, the contribution ratio of the change was calculated to see how much the maldistribution of obstetricians/gynecologists in each group contributed to the change in the Gini coefficient from 1996 to 2016. Here, the method developed by Seki was used [17, 18]. If the pseudo-Gini coefficient of element j (for example, the females under 40 years group) at time t is $R_t^j$, the contribution of the difference from the pseudo-Gini coefficient $R_0^j$ of element j at the reference time ($\Delta R^j$) can be expressed by the following formula.

$$\Delta R^j = W_t^j \cdot R_t^j - W_0^j \cdot R_0^j$$

With $\Delta R^j / \Delta G$ showing the contribution ratio of the change group j to the change in G.

The protocol of this study was reviewed by the Ethical Committee of Toho University School of Medicine, and the committee confirmed that no approval is necessary for this kind of study in Japan because of the anonymous nature of the data (reference number A19034).

## Results

### Current situation of the geographical maldistribution

The present state of the geographical maldistribution is shown in Fig 1. The current geographical maldistribution of obstetricians/gynecologists is extremely high compared to all physicians or pediatricians. The Gini coefficient for obstetricians/gynecologists was rather low in 1996 compared to that of pediatricians. The Gini coefficient for obstetricians/gynecologists was stable for the period between 2002 to 2006, began to rise in 2008, and continued to rise until 2018. It has exceeded 0.4 since 2016. The Gini coefficient for pediatricians gradually declined until 2006, then gradually increased, and has been relatively stable. Although the Gini

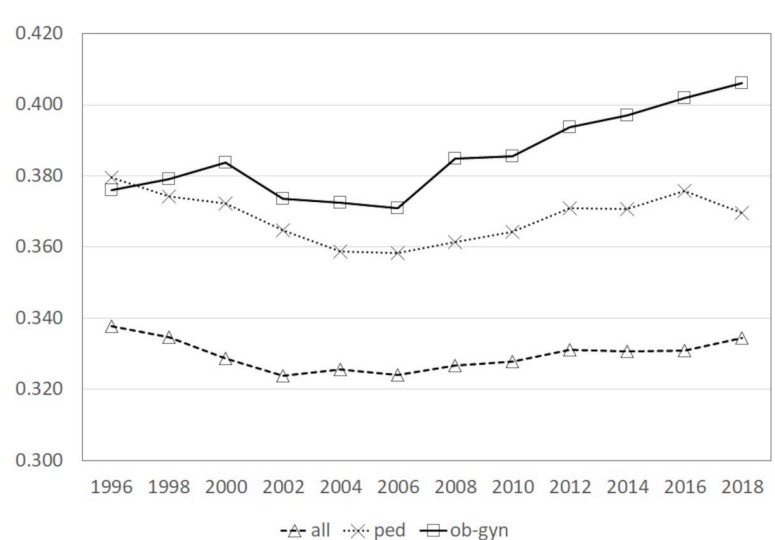

**Fig 1. Gini coefficient of all physicians, pediatricians and obstetricians/gynecologists.** Notes; all: all physicians, ped: pediatricians, and ob-gyn: obstetricians/gynecologists.

coefficient for all physicians is lower compared to that of pediatricians or obstetricians/gynecologists, its trend is similar to that of pediatricians.

## Increase of female obstetrician/gynecologists and geographical maldistribution

Fig 2 shows a time-series comparison of the proportion of females in all physician categories, pediatricians and obstetricians/gynecologists. As at 1996, the proportion of females in obstetricians/gynecologists was only 15.1%, which was not so different from 13.3% of all physicians. The proportion of female pediatricians was high at 28.3% in 1996, and has been increasing moderately since then. However, the proportion of female obstetricians/gynecologists has been rising rapidly since 1996, overtaking that of pediatricians in 2016. In other words, the proportion of young female physicians in obstetricians/gynecologists is high. In fact, in 2018, the proportion of female obstetricians/gynecologists in their 30s was 62.3%, and that of those in their 20s was 65.9%, implying that the number of young female obstetricians/gynecologists is larger than that of male obstetricians/gynecologists.

Table 1 shows how the proportion of females, which has been rising rapidly in recent years, affects the maldistribution of obstetricians/gynecologists. Obstetricians/gynecologists are divided into four groups based on age and gender as described above. The contribution ratio is the percentage of the weighted pseudo-Gini coefficient, which is explained by each group, and adding the percentages of the four groups together gives 100%. The contribution ratios of males under 40 years and males aged 40 years and above have been decreasing, whereas the contribution ratios of females under 40 years and females aged 40 years and above have been increasing. The change in the contribution ratio reflects the change of proportion of male and female physicians. With increasing number of female physicians, the contribution ratio of female physicians rises, and *vice versa*. The maldistribution of obstetricians/gynecologists, which is shown by an increase in the Gini coefficient, has been worsening. This may be attributed to the increase in the proportion of female physicians.

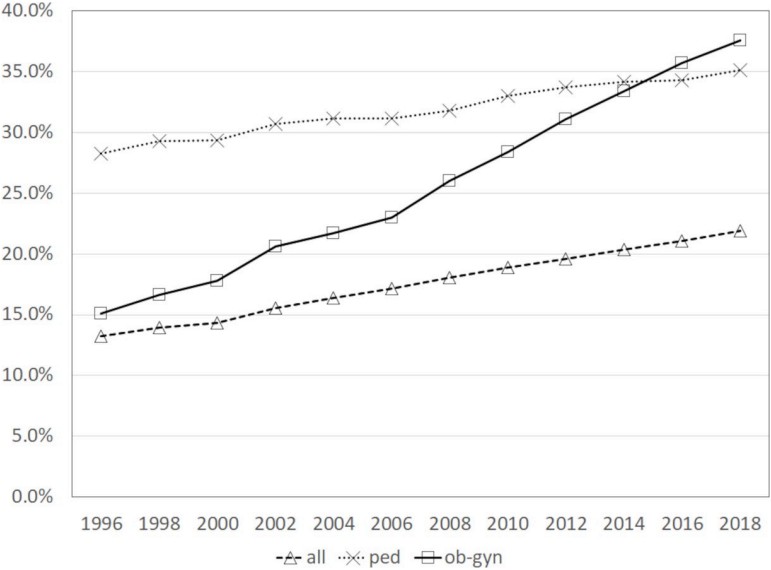

**Fig 2. The percentage of females in all physicians, pediatricians and obstetricians/gynecologists.** Notes; all: all physicians, ped: pediatricians, and ob-gyn: obstetricians/gynecologists.

**Table 1. Contribution ratio to Gini coefficient in each year (the number of physicians).**

|  | 1996 | 1998 | 2000 | 2002 | 2004 | 2006 | 2008 | 2010 | 2012 | 2014 | 2016 |
|---|---|---|---|---|---|---|---|---|---|---|---|
| male <40 y. o. | 19.2% | 18.4% | 16.8% | 15.6% | 13.9% | 11.7% | 10.7% | 10.2% | 10.1% | 10.3% | 9.6% |
|  | (2381) | (2244) | (2028) | (1832) | (1658) | (1584) | (1506) | (1436) | (1446) | (1360) | (1366) |
| female < 40 y. o. | 10.8% | 12.4% | 13.7% | 16.5% | 16.7% | 15.7% | 17.7% | 17.9% | 18.7% | 18.4% | 17.7% |
|  | (758) | (912) | (983) | (1147) | (1318) | (1532) | (1757) | (1936) | (2178) | (2252) | (2287) |
| male ≥ 40 y. o. | 62.0% | 60.8% | 60.9% | 58.7% | 58.2% | 59.7% | 56.6% | 54.6% | 51.8% | 49.7% | 47.9% |
|  | (6866) | (6820) | (6840) | (6677) | (6518) | (6712) | (6597) | (6584) | (6443) | (6391) | (6229) |
| female ≥ 40 y. o. | 8.0% | 8.5% | 8.6% | 9.2% | 11.1% | 12.9% | 15.0% | 17.4% | 19.3% | 21.6% | 24.8% |
|  | (691) | (712) | (684) | (769) | (806) | (923) | (1040) | (1204) | (1372) | (1587) | (1881) |
| Gini Coefficient | 0.376 | 0.379 | 0.384 | 0.374 | 0.373 | 0.371 | 0.385 | 0.386 | 0.394 | 0.397 | 0.402 |

Table 2 shows the contribution ratio of change of each group to change of the Gini coefficient from 1996 to 2016. The contribution ratios were -83.8% for males under 40 years, 85.6% for females under 40 years, -90.6% for males aged 40 years and above, and 188.9% for females aged 40 years and above. Male groups rather contributed to a decrease in the Gini coefficient in 20 years, while female groups contributed to an increase.

Fig 3 shows a time series of the geographical maldistribution in four groups measured by the Gini coefficient. The Gini coefficient for males and females under 40 years was lower than that of those aged 40 years and above. The Gini coefficients for males and females aged 40 years and above were not considerably different, however, the Gini coefficient for females under 40 years was considerably higher than that of males under 40 years. In addition, the Gini coefficient for males under 40 years and males aged 40 years and above tended to increase, whereas the Gini coefficient for females in the same categories tended to decrease in time series. This indicates that the geographical maldistribution is worse within the groups of female obstetricians/gynecologists compared to their male counterparts, but the geographical maldistribution of the former is improving while that of the latter is worsening.

## Discussion

The results of this study showed that the maldistribution of obstetricians/gynecologists was worse compared to that of other physicians or pediatricians. An increase in the number of female obstetricians/gynecologists have potentially contributed to this since female physicians tended to be more unevenly distributed compared to their male counterparts. This may also be related to the limited number of places available for female physicians to work while maintaining their work-life balances. Therefore, the Gini coefficient of females aged 40 years and above was considerably higher than that of males aged 40 years and above. The contribution ratios of female obstetricians/gynecologists have been increasing for those under 40 years and 40 years old and above. The contribution ratio is defined as the ratio of the pseudo-Gini coefficient multiplied by the weight of the group to the whole Gini coefficient. Therefore, if the weight of the group, which originally has a high maldistribution, increases, the contribution ration also increases. It is also due to this weight change that the female physician groups made a positive contribution and the male physician groups made a negative contribution to the increase in

**Table 2. Contribution ratio of each group to change of Gini coefficient (1996→2016).**

| male <40 y. o. | female < 40 y. o. | male ≥ 40 y. o. | female ≥ 40 y. o. |
|---|---|---|---|
| -83.8% | 85.6% | -90.6% | 188.9% |

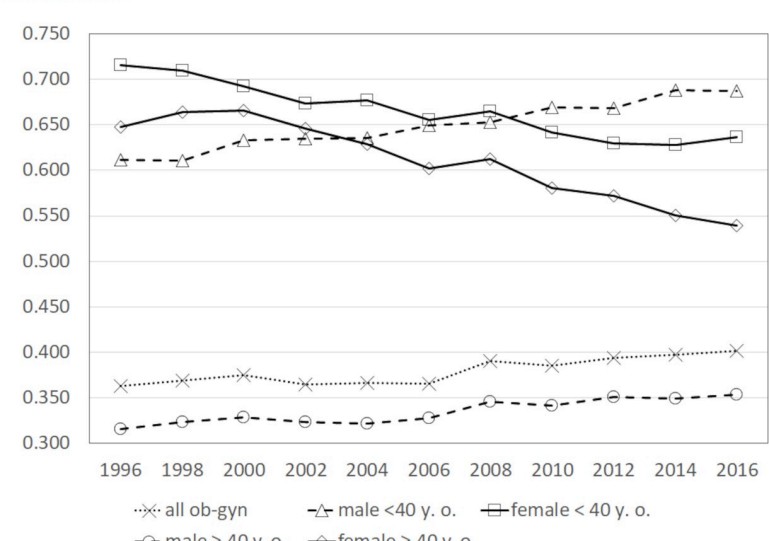

**Fig 3. Trend of Gini coefficient by sex-age groups.** Notes; ob-gyn: obstetricians/gynecologists.

the Gini coefficient. However, looking at the changes in the Gini coefficient for each group, it can be observed that the Gini coefficient continued to decrease in the female groups and did not decrease in the male groups. The maldistribution of under-40-year-old males was worse compared to that of the female group of the same age bracket. This may due to an improvement in the working environment for female physicians and an increase of the areas in which they can work, or an increase in the number of female physicians who have chosen to work in previously unselected hospitals. However, this notion needs to be investigated.

The deterioration of the geographical maldistribution of obstetricians/gynecologists may be due to an increase in the proportion of female obstetricians/gynecologists. However, an improvement in the current working environment for female physicians will help resolve the geographical maldistribution in all obstetricians/gynecologists by improving the geographical maldistribution within female physicians. It is true that the proportion of female obstetricians/gynecologists is increasing, and it is difficult and should not intervene in that situation. Previous studies reported that more female medical students tend to prefer obstetrics and gynecology compared to male medical students [19]. However, the high geographical maldistribution among female obstetricians/gynecologists might reflect the fact that female physicians were not expected to work as obstetricians/gynecologists. An improvement in the working environment as well as stimulating the interests of more numbers of medical students in obstetrics and gynecology can help address the geographical distribution of this class of specialist physicians.

The major limitation of this study is that we could not analyze/evaluate the influence (if any) of the specialist qualification system. The specialist system has been recently reformed, with a new instituted in 2018. In the new system, the third-party organization collectively manages specialist qualifications, which were hitherto accredited by Academic Societies. The geographical maldistribution of training facilities may deteriorate the geographical maldistribution of physicians. Hence, it would be necessary to refer to this new specialist system in discussing the maldistribution of physicians. However, we think that it may take some time for the system to appear in statistical data due to its newness. In addition, the worsening uneven distribution of obstetricians/gynecologists is considered to be largely influenced by the

government and Academic Societies that have promoted the selection and concentration of obstetric hospitals [20, 21]. If the obstetrics/gynecology workforce continues to decline, it will force individual doctors in the facilities that handle delivery to overwork, and concurrently lead to a decline in the quality of obstetric care. Therefore, the government and the Japan Society of Obstetrics and Gynecology have promoted the selection and concentration of obstetric hospitals. As a result, the uneven distribution of regions in each municipality may have increased. Furthermore, it is considered that the uneven distribution of obstetricians and obstetricians and gynecologists has further expanded due to the increase in the proportion of female doctors in parallel with this movement. The concentration is aimed at improving the working environment and ensuring a sustainable perinatal medical system, but it may also worsen access to delivery facilities in rural areas. It will become even more important to develop the working environment to enable the complete utilization of the skills of increasing number of female physicians.

## Conclusion

The deterioration of geographical maldistribution of obstetricians/gynecologists in Japan was influenced by the increase in the proportion of female physicians who rarely work in rural areas. However, an improvement in the current working environment for female physicians will help resolve the geographical maldistribution in all obstetricians/gynecologists by improving the geographical maldistribution within female physicians.

## Author Contributions

**Conceptualization:** Kunichika Matsumoto, Eijiro Hayata.

**Data curation:** Kunichika Matsumoto.

**Formal analysis:** Kunichika Matsumoto.

**Funding acquisition:** Kunichika Matsumoto.

**Investigation:** Kunichika Matsumoto.

**Methodology:** Kunichika Matsumoto.

**Project administration:** Kunichika Matsumoto, Kanako Seto.

**Resources:** Kunichika Matsumoto.

**Supervision:** Tomonori Hasegawa.

**Visualization:** Kunichika Matsumoto.

**Writing – original draft:** Kunichika Matsumoto.

**Writing – review & editing:** Kanako Seto, Eijiro Hayata, Shigeru Fujita, Yosuke Hatakeyama, Ryo Onishi, Tomonori Hasegawa.

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
