## [Decision Letter · Decision Letter 0]

7 Dec 2020

PONE-D-20-32763

The Geographical Maldistribution of Obstetricians and Gynecologists in Japan

PLOS ONE

Dear Dr. Hasegawa,

Thank you for submitting your manuscript to PLOS ONE. After careful consideration, we feel that it has merit but does not fully meet PLOS ONE’s publication criteria as it currently stands. Therefore, we invite you to submit a revised version of the manuscript that addresses the points raised during the review process.

We look forward to receiving your revised manuscript.

Kind regards,

Antonio Simone Laganà, M.D., Ph.D.

Academic Editor

PLOS ONE

Journal Requirements:

2.) We note that you have indicated that data from this study are available upon request. PLOS only allows data to be available upon request if there are legal or ethical restrictions on sharing data publicly. For information on unacceptable data access restrictions, please see http://journals.plos.org/plosone/s/data-availability#loc-unacceptable-data-access-restrictions.

Additional Editor Comments (if provided):

The reviewer have expressed positive comments regarding your article, raising only few concerns. Considering this point, I invite authors to perform the required minor revisions.

Reviewers' comments:

Reviewer's Responses to Questions

**Comments to the Author**

1. Is the manuscript technically sound, and do the data support the conclusions?

Reviewer #1: Yes

2. Has the statistical analysis been performed appropriately and rigorously? 

Reviewer #1: Yes

3. Have the authors made all data underlying the findings in their manuscript fully available?

Reviewer #1: Yes

4. Is the manuscript presented in an intelligible fashion and written in standard English?

Reviewer #1: Yes

5. Review Comments to the Author

Reviewer #1: Thank you for giving the opportunity to review this interesting paper. This paper showed the geographic distribution of OBGYN physicians in Japan sorted by gender and age. Although the OBGYN workforce and distribution has been treated in some of past papers, the viewpoint of this paper is quite unique and new. The implications of the results of this study are of utmost importance in Japanese health policy. I have some minor points which can be raised as issues.

L101 Do you mean you were permitted by the Ministry of Health to use the individualized data for the research purpose? If so, you need to clearly state so and add the permission number issued by the Ministry.

L171 The word “significant” should be avoided because statistical test for the difference is not conducted (if you done, the results would've been much more persuasive..).

L208-215 It is utterly unscientific that you don’t mention the huge gap of Gini coefficient between male >40 and female >40 yo. It looks like “the elephant in the room”, which is far more important than the trend of the coefficient itself.

L220 The absolute value of coefficient itself (0.400) is not so meaningful because it changes substantially depending on the variables used and geographic unit of analysis.

L220 “might” should be “potentially”

L260- You might consider the effect of the nature of OBGY practice on geographic distribution. (Tohoku J Exp Med 2020 May;251(1):1-8. doi: 10.1620/tjem.251.1.) Also please pay attention to the “selection and concentration” of obstetricians into urban high-volume hospitals, which is proposed and facilitated forcefully both by the government and the Japan OBGY Association. (Int J Health Geogr. 2016 Jan 22;15:4. doi: 10.1186/s12942-016-0035-y; J Obstet Gynaecol Res. 2015 Jun;41(6):919-25. doi: 10.1111/jog.12663. )

6. PLOS authors have the option to publish the peer review history of their article (what does this mean?). If published, this will include your full peer review and any attached files.

Reviewer #1: No

---

## [Author Response · Author response to Decision Letter 0]

21 Dec 2020

We wish to express our appreciation to the Reviewer for his or her insightful comments, which have helped us significantly to the paper. 

Comment: 

L101 Do you mean you were permitted by the Ministry of Health to use the individualized data for the research purpose? If so, you need to clearly state so and add the permission number issued by the Ministry.

Response:

Yes. We added the following sentence.

L102

Under the permission of the Ministry of Health, Labour and Welfare (Seito-0408-1), we obtained anonymized individual data from this census.

Comment:

L171 The word “significant” should be avoided because statistical test for the difference is not conducted (if you done, the results would've been much more persuasive..).

Response:

We deleted the word “significant”.

L173

The Gini coefficient for obstetricians/gynecologists was stable for the period between 2002 to 2006, began to rise in 2008, and continued to rise until 2018.

Comment:

L208-215 It is utterly unscientific that you don’t mention the huge gap of Gini coefficient between male >40 and female >40 yo. It looks like “the elephant in the room”, which is far more important than the trend of the coefficient itself.

Response:

We added sentences in Result section and Discussion section.

L214

The Gini coefficients for males and females aged 40 years and above were not considerably different, however, the Gini coefficient for females under 40 years was considerably higher than that of males under 40 years. In addition,

L230

Therefore, the Gini coefficient of females aged 40 years and above was considerably higher than that of males aged 40 years and above.

Comment:

L220 The absolute value of coefficient itself (0.400) is not so meaningful because it changes substantially depending on the variables used and geographic unit of analysis.

Response:

We also know the magnitude of the Gini coefficient itself is not so meaningful. So, we deleted the absolute value.

L225

The results of this study showed that the maldistribution of obstetricians/gynecologists was worse compared to that of other physicians or pediatricians.

Comment:

L220 “might” should be “potentially”

Response:

Thank you. We changed the sentence.

L226

An increase in the number of female obstetricians/gynecologists have potentially contributed to this since female physicians tended to be more unevenly distributed compared to their male counterparts.

Comment:

L260- You might consider the effect of the nature of OBGY practice on geographic distribution. (Tohoku J Exp Med 2020 May;251(1):1-8. doi: 10.1620/tjem.251.1.) Also please pay attention to the “selection and concentration” of obstetricians into urban high-volume hospitals, which is proposed and facilitated forcefully both by the government and the Japan OBGY Association. (Int J Health Geogr. 2016 Jan 22;15:4. doi: 10.1186/s12942-016-0035-y; J Obstet Gynaecol Res. 2015 Jun;41(6):919-25. doi: 10.1111/jog.12663. )

Response:

We added the following sentences and 2 more references.

L269

In addition, the worsening uneven distribution of obstetricians/gynecologists is considered to be largely influenced by the government and Academic Societies that have promoted the selection and concentration of obstetric hospitals [21, 22]. If the obstetrics/gynecology workforce continues to decline, it will force individual doctors in the facilities that handle delivery to overwork, and concurrently lead to a decline in the quality of obstetric care. Therefore, the government and the Japan Society of Obstetrics and Gynecology have promoted the selection and concentration of obstetric hospitals. As a result, the uneven distribution of regions in each municipality may have increased. Furthermore, it is considered that the uneven distribution of obstetricians and obstetricians and gynecologists has further expanded due to the increase in the proportion of female doctors in parallel with this movement. The concentration is aimed at improving the working environment and ensuring a sustainable perinatal medical system, but it may also worsen access to delivery facilities in rural areas. It will become even more important to develop the working environment to enable the complete utilization of the skills of increasing number of female physicians.

---

## [Decision Letter · Decision Letter 1]

30 Dec 2020

The Geographical Maldistribution of Obstetricians and Gynecologists in Japan

PONE-D-20-32763R1

Dear Dr. Hasegawa,

We’re pleased to inform you that your manuscript has been judged scientifically suitable for publication and will be formally accepted for publication once it meets all outstanding technical requirements.

Kind regards,

Antonio Simone Laganà, M.D., Ph.D.

Academic Editor

PLOS ONE

Additional Editor Comments (optional):

Authors performed the required corrections, which were positively evaluated by the reviewers. I am pleased to accept this paper for publication.

Reviewers' comments:

Reviewer's Responses to Questions

**Comments to the Author**

1. If the authors have adequately addressed your comments raised in a previous round of review and you feel that this manuscript is now acceptable for publication, you may indicate that here to bypass the “Comments to the Author” section, enter your conflict of interest statement in the “Confidential to Editor” section, and submit your "Accept" recommendation.

Reviewer #1: All comments have been addressed

2. Is the manuscript technically sound, and do the data support the conclusions?

Reviewer #1: (No Response)

3. Has the statistical analysis been performed appropriately and rigorously? 

Reviewer #1: (No Response)

4. Have the authors made all data underlying the findings in their manuscript fully available?

Reviewer #1: (No Response)

5. Is the manuscript presented in an intelligible fashion and written in standard English?

Reviewer #1: (No Response)

6. Review Comments to the Author

Reviewer #1: (No Response)

7. PLOS authors have the option to publish the peer review history of their article (what does this mean?). If published, this will include your full peer review and any attached files.

Reviewer #1: No

---

## [Editor Report · Acceptance letter]

4 Jan 2021

PONE-D-20-32763R1 

The Geographical Maldistribution of Obstetricians and Gynecologists in Japan 

Dear Dr. Hasegawa:

I'm pleased to inform you that your manuscript has been deemed suitable for publication in PLOS ONE. Congratulations! Your manuscript is now with our production department. 

Kind regards, 

on behalf of

Dr. Antonio Simone Laganà 

Academic Editor

PLOS ONE